# Spectral Operator Methods for Learning Coherent Temporal Representations in Cellular Signaling Dynamics

## Abstract

We present a novel operator-based framework for learning coherent temporal representations of cellular dynamics from live-cell imaging data. Recognizing the inherent stochasticity and measurement limitations in biological systems, our approach shifts the focus from predicting exact trajectories to characterizing key dynamical properties that shape cellular behaviors at the population level. By leveraging spectral analysis of the Koopman operator and smoothing via Markov semigroups of kernel integral operators, we identify near-resonant patterns and transient coherent structures that persist across different experimental conditions. This methodology effectively captures fundamental dynamics, providing insights into mechanisms of heterogeneous cell responses without the need to model precise transformation laws. We demonstrate the efficacy of our framework on a dataset of retinal pigment epithelial cells with an inducible oncogene, revealing conserved dynamical patterns across varying levels of ERK inhibition. Our work offers interpretable learned representations, even with limited and noisy single-cell-resolved recordings, advancing machine learning for dynamical systems and opening new avenues for understanding and predicting cellular behavior in response to external stimuli.

## 1 Introduction

Understanding complex dynamical behaviors of cellular signaling networks remains a fundamental challenge in computational biology and machine learning Ideker et al. (2001). Unlike engineered systems with deterministic functions and precise equations of motion, cellular dynamics emerge from the interactions of small numbers of molecules whose combinatorial complexity leads to inherent stochasticity Eldar & Elowitz (2010); Altschuler & Wu (2010). More precisely, operating far from thermodynamic limits where large numbers would average out fluctuations, these subcellular systems exhibit pronounced stochastic effects - from spontaneous switching between cellular states to heterogeneous responses to perturbations Elowitz et al. (2002); Spencer et al. (2009).

Factors such as interacting signaling pathways, varying mRNA half-lives, and fluctuating environments contribute to intrinsic stochasticity within genetically identical (isogenic) cell populations Eldar & Elowitz (2010); Altschuler & Wu (2010). Rather than viewing this stochasticity as experimental noise to be filtered out, we recognize it as a fundamental feature that both enables cellular decision-making and induces signatures for identifying robust dynamical patterns Purvis & Lahav (2013); Levine et al. (2013). A striking example is how cells achieve coordination among groups of co-regulated genes (regulons) through noise-driven mechanisms Eldar & Elowitz (2010). These mechanisms operate across multiple scales, from molecular fluctuations that trigger gene expression switches to population-level coordination of cellular states Elowitz et al. (2002). Capturing these complex dynamics is further complicated by limitations in measurement technologies. Traditional high-throughput single-cell technologies enable rapid collection of distributions across diverse conditions Lin et al. (2015; 2016) but lack temporal pairing between cells Weinreb et al. (2018). While live-cell imaging provides time-resolved measurements Cutrale et al. (2017), it is limited to tracking only a few variables simultaneously due to technical constraints Stewart et al. (2016). Consequently, analyzing cellular dynamics from live-cell imaging presents significant challenges due to both intrinsic stochastic fluctuations and extrinsic heterogeneity between cells. This heterogeneity, which

single-cell analysis aims to uncover, makes it particularly difficult to distinguish between transient behaviors and to build predictive models. Current approaches often average out cell-to-cell variations, obscuring the very heterogeneity that single cell data is designed to uncover and only provide phenomenological descriptions without mechanistic insights Snijder & Pelkmans (2011).

While several methods have been developed for analyzing live-cell data, none fully addresses the challenges of modeling cellular dynamics. CODEX (Jacques et al., 2021) employs convolutional neural networks for pattern recognition in time-series data. However, it treats cellular trajectories as static patterns for classification rather than as evolving dynamical systems. While effective at identifying recurring motifs, CODEX does not explicitly model the underlying dynamics or stochastic processes, requires large training datasets, and produces models that are challenging to interpret mechanistically. Functional principal component analysis (fPCA) has been applied to analyze variability in live-cell imaging data (Sampattavanich et al., 2018), particularly for studying temporal changes in molecular concentrations between nucleus and cytoplasm. While fPCA effectively decomposes trajectories into orthogonal modes capturing dominant patterns, its optimization for variance explained rather than dynamical features means these components may not correspond to meaningful biological processes. Moreover, fPCA cannot predict beyond the observed time window as it does not model the generating system, and manual selection of components can introduce bias.

More established tools in system identification have attempted to address similar limitations. Stable linear dynamical systems (LDS) (Boots, 2009) and its extensions for high-dimensional settings (Chen et al., 2017) provide computationally tractable methods through reduced-rank approximations. However, these methods make restrictive assumptions that limit their ability to capture complex nonlinear dynamics. Their linear evolution assumptions cannot capture nonlinear interactions such as transitions present in biological data, their Gaussian noise models may not reflect true stochastic processes, and their dimensionality reduction can discard important dynamical information. In contrast, our operator-based approach using the Koopman framework explicitly models system evolution without linearity assumptions. By lifting nonlinear dynamics into a linear framework through the action on observables, and regularizing through Markov semigroups, we obtain a mathematically rigorous method with provable convergence properties. Rather than relying on predetermined dimensionality reduction, our method adaptively determines relevant modes through spectral analysis of the regularized operator. This allows us to capture rich nonlinear behaviors while maintaining computational tractability and providing theoretical guarantees about convergence to the true dynamics - key features lacking in current approaches.

Operator-theoretic approaches combined with data-driven learning offer a promising alternative by identifying patterns directly from single-cell measurements while preserving the essential heterogeneity that drives cellular decision-making Das & Giannakis (2019); Mezić (2005). Rather than attempting to learn all behaviors, most of which are unpredictable, we focus on identifying coherent temporal patterns that persist for finite times–analogous to studying coherent structures in turbulent flows Mezić (2013). The Koopman operator approach is particularly promising in this context. By representing dynamics through the action on functions, e.g. fluorescent readouts of protein levels, and through spectral analysis, we can identify near-resonances that shape transient responses to perturbations like drug treatments. Our approach combines and extends several powerful concepts:

1. The Koopman operator framework, which enables study of nonlinear dynamics through linear methods while naturally handling stochastic effects Mezić (2005); Das & Giannakis (2019)

2. Kernel methods that transform complex data into spaces where dynamical patterns become apparent Berry et al. (2015)

3. Regularization techniques via Markov semigroups that make infinite-dimensional problems computationally tractable while preserving biologically relevant features Giannakis (2015)

We demonstrate our framework's effectiveness using live-cell imaging data from cells under various perturbations Chen et al. (2023), showing how it captures coherent temporal patterns that persist despite high variability while highlighting condition-specific dynamics.

## 2 DYNAMICAL SYSTEM REPRESENTATION

In this section, we present our *Operator-Based Dynamics Framework* for learning coherent temporal representations from live-cell trajectory data. We define coherent temporal patterns as robust, recurring, and interpretable structures in the time evolution of the system that persist across temporal scales and capture the intrinsic dynamical organization, including periodic cycles, stable trends, attracting sets, and variability patterns. The results demonstrate that these coherent patterns substantially improve the transferability and generalization capabilities of the models across diverse datasets. We formulate the cellular signaling response as a dynamical system with state space $\mathbb{X} \subseteq \mathbb{R}^d$ and flow map $\Phi : \mathbb{X} \times \mathbb{T} \to \mathbb{X}$, where $\mathbb{T} \subseteq \mathbb{R}$ denotes time. The flow map $\Phi(x, \Delta t) = \Phi^{\Delta t}(x)$ characterizes the evolution of an initial state $x \in \mathbb{X}$ over time interval $\Delta t \in \mathbb{T}$, describing the deterministic dynamics of the system. To account for inherent uncertainties arising from molecular noise and environmental fluctuations, we extend beyond deterministic dynamics to incorporate stochastic behavior. We represent the system state at time $t$ as a random variable $X_t$ with an associated probability distribution over $\mathbb{X}$. This probabilistic framework enables characterization of the system evolution through state transitions over time.

To model the probabilistic evolution of the system, we introduce the *transition density function* $p_{\Delta t} : \mathbb{X} \times \mathbb{X} \to [0, \infty)$, which describes the probability density of transitioning from state $x \in \mathbb{X}$ at time $t$ to state $y \in \mathbb{X}$ at time $t + \Delta t$. For a measurable subset $\mathbb{A} \subseteq \mathbb{X}$, the probability of the system transitioning from state $x$ to $\mathbb{A}$ over time $\Delta t$ is given by:

$$\mathbb{P}[\Phi^{\Delta t}(\mathbf{x}_t) \in \mathbb{A} \mid \mathbf{x}_t = x] = \int_{\mathbb{A}} p_{\Delta t}(x, y) \, \mu(dy), \tag{1}$$

where $\mu$ is a measure on $\mathbb{X}$, typically the Lebesgue measure when $\mathbb{X}$ is a subset of $\mathbb{R}^d$. The probabilistic evolution of densities over time can be described using the *Perron-Frobenius operator* (also known as the *transfer operator*) $\mathcal{P}^{\Delta t}$. This operator acts on functions $f \in L^1(\mathbb{X}, \mu)$ and describes how a probability density evolves under the dynamics induced by $\Phi^{\Delta t}$. Formally, for a measure space $(\mathbb{X}, \mathcal{B}, \mu)$, where $\mathcal{B}$ is the Borel sigma-algebra on $\mathbb{X}$, and for any measurable subset $\mathbb{A} \in \mathcal{B}$, the Perron-Frobenius operator $\mathcal{P}^{\Delta t} : L^1(\mathbb{X}, \mu) \to L^1(\mathbb{X}, \mu)$ satisfies:

$$\int_{\mathbb{A}} (\mathcal{P}^{\Delta t} f)(x) \, \mu(dx) = \int_{\Phi^{-\Delta t}(\mathbb{A})} f(x) \, \mu(dx). \tag{2}$$

This equation states that the total probability mass in set $\mathbb{A}$ at time $t + \Delta t$ is equal to the total probability mass in the pre-image $\Phi^{-\Delta t}(\mathbb{A})$ at time $t$, where $\Phi^{-\Delta t}$ denotes the backward flow over time $\Delta t$. The operator $\mathcal{P}^{\Delta t}$ is linear and preserves total probability, i.e., if $f$ is a probability density function, so is $\mathcal{P}^{\Delta t} f$. Alternatively, when the transition density function $p_{\Delta t}(x, y)$ exists, the action of the Perron–Frobenius operator can be expressed as:

$$(\mathcal{P}^{\Delta t} f)(y) = \int_{\mathbb{X}} p_{\Delta t}(x, y) f(x) \mu(dx). \tag{3}$$

**Koopman Operator:** Complementary to the Perron–Frobenius operator, which describes the evolution of densities, the *Koopman operator* $\mathcal{K}^{\Delta t}$ acts on observables (functions of the state) and captures how these observables evolve under the dynamics. Specifically, for an observable function $g \in L^\infty(\mathbb{X}, \mu)$, the Koopman operator $\mathcal{K}^{\Delta t} : L^\infty(\mathbb{X}, \mu) \to L^\infty(\mathbb{X}, \mu)$ is defined as:

$$(\mathcal{K}^{\Delta t} g)(x) = \mathbb{E}[g(\Phi(\mathbf{x}_t)) \mid \mathbf{x}_t = x] = \int_{\mathbb{X}} g(y) p_{\Delta t}(x, y) \, \mu(dy). \tag{4}$$

The Koopman operator is also linear, even if the underlying dynamics are nonlinear and stochastic. It provides a linear representation of the evolution of observables under the dynamics. Moreover, the Koopman operator is the adjoint of the Perron-Frobenius operator with respect to the inner product in $L^2(\mathbb{X}, \mu)$, i.e., for $f \in L^1(\mathbb{X}, \mu)$ and $g \in L^\infty(\mathbb{X}, \mu)$,

$$\int_{\mathbb{X}} (\mathcal{K}^{\Delta t} g)(x) f(x) \, \mu(dx) = \int_{\mathbb{X}} g(x) (\mathcal{P}^{\Delta t} f)(x) \, \mu(dx). \tag{5}$$

This duality allows us to study the dynamics either through the evolution of densities (Perron-Frobenius operator) or through the evolution of observables (Koopman operator).

## 2.1 Spectral Analysis of the Koopman Operator

As a linear operator, the Koopman operator $\mathcal{K}^{\Delta t}$ can be decomposed into its eigenfunctions and eigenvalues. Specifically, we seek eigenfunctions $\phi_k \in L^\infty(\mathbb{X}, \mu)$ and corresponding eigenvalues $\lambda_k \in \mathbb{C}$ satisfying:

$$\mathcal{K}^{\Delta t}\phi_k = \lambda_k\phi_k. \tag{6}$$

These eigenfunctions represent modes of the system that evolve linearly over time. By approximating these eigenfunctions and eigenvalues, we can decompose complex, nonlinear, and stochastic dynamics into a superposition of simpler, linear modes.

**Pseudospectra of the Koopman:** Given the stochastic and transient nature of cellular dynamics and the limitations in predicting exact trajectories, we adopt a *pseudospectrum approach* to identify coherent dynamical patterns that are robust to perturbations and uncertainties. The $\epsilon$-pseudospectrum of the Koopman operator $\mathcal{K}^{\Delta t}$, denoted by $\sigma_\epsilon(\mathcal{K}^{\Delta t})$, consists of all complex numbers $\lambda \in \mathbb{C}$ for which the resolvent norm is large:

$$\sigma_\epsilon(\mathcal{K}^{\Delta t}) = \left\{ \lambda \in \mathbb{C} \;\middle|\; \left\| \left(\mathcal{K}^{\Delta t} - \lambda I\right)^{-1} \right\| \geq \frac{1}{\epsilon} \right\}. \tag{7}$$

However, working directly with resolvents can be computationally challenging (Sharma et al., 2016; Giannakis & Valva, 2024; Colbrook & Townsend, 2021; Colbrook et al., 2023). Therefore, in the subsequent sections, we adopt an alternative approach to analyze finite-time dynamics and transient behaviors by employing a method based on smoothing via a Markov semigroup of kernel integral operators (Valva & Giannakis, 2024). While this approach may not yield the exact pseudospectrum due to the regularization of the Koopman operator, the eigenfunctions of the smoothed operator still represent coherent temporal patterns that persist over finite timescales. While this approach yields a different spectrum from the original Koopman operator or its pseudospectrum, it effectively captures near-resonant behaviors and coherent transient patterns in the dynamics, similar to the pseudospectrum approach.

**Identification of Coherent Dynamical Patterns** Approximate eigenfunctions obtained from the smoothing method represent coherent temporal patterns in the cellular dynamics that persist over finite timescales. These patterns evolve nearly linearly and can be used to decompose the complex dynamics into a sum of simpler, predictable components. For an approximate eigenfunction $\phi_j$, the evolution under the Koopman operator satisfies:

$$\mathcal{K}^{n\Delta t}g \approx \lambda^n g, \tag{8}$$

for $n \in \mathbb{N}$. This approximation holds over finite timescales where the patterns remain coherent. By expressing observables as linear combinations of these approximate eigenfunctions, we obtain a spectral decomposition of the dynamics:

$$g(x) = \sum_j \phi_j(x)c_j, \tag{9}$$

where $\phi_j$ are the approximate eigenfunctions and $c_j$ are coefficients. The evolution of $g$ is then approximated by:

$$\mathcal{K}^{n\Delta t}g(x) \approx \sum_j \lambda_j^n \phi_j(x)c_j. \tag{10}$$

This decomposition allows us to capture the dominant temporal patterns in the data, even when exact trajectory prediction is impossible. The approximate eigenfunctions $\phi_j$ correspond to modes that represent collective behaviors of the system, providing insights into the mechanisms underlying cellular responses.

## 2.2 Learning the Spectral Components of the Dynamics

To extract coherent temporal patterns from live-cell trajectory data, we employ a data-driven approach to approximate the Koopman operator. Before detailing the approximation method, we first describe the data and the experimental conditions under which it was collected.

**Experimental Conditions**    Let $\{C_k\}_{k=1}^K$ denote the different experimental conditions under which live-cell imaging data were collected. Each condition $C_k$ represents a specific perturbation or treatment applied to the cells, such as varying doses of an inhibitor or other perturbations. For each condition, we observe $N_k$ cell trajectories, where each trajectory consists of time-series measurements over $T$ time points. The measurements are denoted by $\{x_t^{(i,k)}\}_{t=0}^T$ for the $i$-th cell in condition $C_k$, where $x_t^{(i,k)} \in \mathbb{R}^d$ represents the state vector of observable quantities (e.g., fluorescence intensities corresponding to signaling molecule activities) at time $t$.

**Delay-Coordinate Embedding**    To capture the underlying dynamics of the system and obtain a data-informed geometry suitable for constructing a Markov operator, we employ delay-coordinate embedding. This method reconstructs the phase space of the dynamical system using time-delayed observations of the measured variables, effectively unfolding the dynamics into a higher-dimensional space (Takens, 1996). For each trajectory, we construct a delay-coordinate map $F_Q : \mathbb{X} \to \mathbb{R}^{Qd}$ defined by

$$F_Q(x_t) = \left[x_t^\top, x_{t-\Delta t}^\top, x_{t-2\Delta t}^\top, \ldots, x_{t-(Q-1)\Delta t}^\top\right]^\top, \tag{11}$$

where $Q$ is the number of delays and $\Delta t$ is the sampling interval. The delay-coordinate embedding captures the temporal structure of the data, allowing us to reconstruct the dynamics even when only a few variables are measured.

**Kernel Function and Integral Operator**    Using the embedded data, we define a kernel function $k : \mathbb{R}^{Qd} \times \mathbb{R}^{Qd} \to \mathbb{R}_+$ to quantify the similarity between points. We employ a self-tuning kernel that adapts to the local data density (Berry & Harlim, 2016):

$$k(x, y) = \exp\left(-\frac{\|x - y\|^2}{\sigma(x)\sigma(y)}\right), \tag{12}$$

where $\| \cdot \|$ denotes the Euclidean norm, and $\sigma(x)$ is a local bandwidth parameter. This kernel captures local structures while being robust to variations in data density.

We then construct an integral operator $K$ acting on functions $f : \mathbb{R}^{Qd} \to \mathbb{R}$:

$$(Kf)(x) = \int_{\mathbb{R}^{Qd}} k(x, y)f(y) \, d\mu(y), \tag{13}$$

where $\mu$ is the empirical measure derived from the data.

**Markov Operator and Eigenvalue Problem**    To analyze the dynamics in probability spaces, we normalize the kernel to construct a Markov operator. The normalization involves computing the degree function

$$d(x) = \int_{\mathbb{R}^{Qd}} k(x, y) \, d\mu(y), \tag{14}$$

and then normalizing the kernel:

$$\tilde{k}(x, y) = \frac{k(x, y)}{d(x)}. \tag{15}$$

The normalized kernel defines a Markov operator $P$:

$$(Pf)(x) = \int_{\mathbb{R}^{Qd}} \tilde{k}(x, y)f(y) \, d\mu(y). \tag{16}$$

This Markov operator $P$ forms the basis for constructing the Markov semigroup $P_\tau$, parameterized by $\tau > 0$, which we will use for smoothing in our spectral approximation. In discrete form, for $N$ data points $x_i{}_{i=1}^N$, the Markov matrix $P$ has entries:

$$P_{ij} = \frac{K_{ij}}{\left(\sum_{k=1}^N K_{ik} q_k^{-1/2}\right) q_j^{1/2}}, \quad q_i = \sum_{k=1}^N K_{ik} \tag{17}$$

We compute the eigenvalues $\gamma_j$ and corresponding eigenvectors $\varphi_j$ of $P$ by solving the eigenvalue problem:

$$P\varphi_j = \lambda_j\varphi_j. \tag{18}$$

The eigenvalues are real and satisfy $1 = \gamma_1 \geq \gamma_2 \geq \cdots \geq \gamma_N \geq -1$. The leading eigenvector $\varphi_1$ corresponds to the stationary distribution of the Markov chain. These eigenvectors $\varphi_j$ will serve as the basis for our Galerkin approximation of the smoothed Koopman operator.

**Sparse Representation**   To handle large datasets efficiently, we construct a $k$-nearest neighbor graph to sparsify the kernel matrix. For each data point $x_i$, we connect it to its $k$ nearest neighbors based on the Euclidean distance in the embedded space. The kernel function is then applied only to these neighboring pairs, resulting in a sparse kernel matrix $K$ and, consequently, a sparse Markov matrix $P$. This sparsity reduces computational complexity and storage requirements, making the method scalable to large datasets. While the Markov operator $P$ captures the dynamics of the system, direct spectral analysis may be sensitive to noise and perturbations. To address this, we introduce a smoothing approach using a Markov semigroup, which will be detailed in the following Galerkin approximation section.

**Markov Semigroup for Smoothing**   To enhance the robustness of our spectral analysis to noise and perturbations, we introduce a Markov semigroup $P_\tau$. This semigroup is generated by the Markov operator $P$ and is defined for $\tau \geq 0$ as:

$$P_\tau = e^{\tau(P-I)}, \tag{19}$$

where $I$ is the identity operator. The semigroup satisfies the properties: $P_0 = I$, $P_{\tau_1}P_{\tau_2} = P_{\tau_1+\tau_2}$ for all $\tau_1, \tau_2 \geq 0$ and strongly continuous at 0, i.e. $\lim_{\tau\to 0^+} |P_\tau f - f| = 0$ for all $f$ in the domain of $P$.

The parameter $\tau$ controls the amount of smoothing: as $\tau$ increases, $P_\tau$ becomes increasingly diffusive, smoothing out fine-scale features in the data.

**Galerkin Approach and Smoothing by Markov Semigroup**   To approximate the Koopman operator and its eigenfunctions, we employ a Galerkin method (Rowley et al., 2009; Klus, 2020) incorporating smoothing by a Markov semigroup of kernel integral operators (Valva & Giannakis, 2024). We project the smoothed Koopman operator onto the subspace spanned by the leading $L$ eigenvectors of the Markov operator $P$, denoted as $\{\varphi_j\}_{j=1}^{L}$. The smoothing process is achieved through the application of the Markov semigroup $P_\tau$, parameterized by $\tau > 0$. We approximate the eigenfunctions of the smoothed operator as linear combinations:

$$\phi_\tau = \sum_{j=1}^{L} c_j\varphi_j. \tag{20}$$

The coefficients $c_j$ are determined by enforcing that the action of the smoothed Koopman operator on $\phi_\tau$ is approximated within the chosen subspace. Specifically, we consider the finite-dimensional approximation of the smoothed Koopman operator $\mathbf{K}_\tau$, defined by:

$$\mathbf{K}_\tau = \mathbf{G}^{-1}\mathbf{A}_\tau, \tag{21}$$

where $\mathbf{G}$ is the Gram matrix and $\mathbf{A}_\tau$ is the smoothed covariance matrix, with entries:

$$G_{ij} = \langle\varphi_i, \varphi_j\rangle, \quad A_{\tau,ij} = \langle\varphi_i, P_{\tau/2}\mathcal{K}P_{\tau/2}\varphi_j\rangle. \tag{22}$$

Here, $P_{\tau/2}$ represents the action of the Markov semigroup, which smooths the Koopman operator. The inner product $\langle\cdot,\cdot\rangle$ is approximated using the empirical data as before.

To compute the entries of $\mathbf{A}_\tau$, we approximate the action of the smoothed Koopman operator on the basis functions using the time-series data and the kernel integral operator. Assuming that $x_{n+1}$ follows $x_n$ in the data, we have:

---

**Algorithm 1** Koopman Eigenfunction Approximation

---

**Require:** Time series $\{x_k\} \in \mathbb{R}^d$, delays $Q$, neighbors $k_{nn}$, Markov eigenfunctions $l \leq N$, regularization $\theta \geq 0$, output dim $l' \leq l$

**Ensure:** Koopman eigenvalues $\{\lambda_k\}_{k=1}^{l'} \in \mathbb{C}$, frequencies $\{\omega_k\}_{k=1}^{l'} \in \mathbb{R}$, eigenfunctions $\{\psi_k\}_{k=1}^{l'} \in \mathbb{C}^N$

1: Compute pairwise distances $d_Q^2(x_i, x_j) = \frac{1}{Q} \sum_{k=0}^{Q-1} \| x_{i-k} - x_{j-k} \|^2$
2: Retain $k_{nn}$ nearest neighbors for each point $i$ in set $\mathcal{N}_{k_{nn}}(x_i)$
3: Symmetrize distances by augmenting if $x_i \in \mathcal{N}_{k_{nn}}(x_j)$ but $x_j \notin \mathcal{N}_{k_{nn}}(x_i)$
4: Compute bandwidth $\epsilon(x_i, x_j)$ (Berry & Harlim, 2016)
5: Form kernel matrix $K_{ij} = \exp(-d_Q^2(x_i, x_j)/\epsilon)$
6: Compute normalized matrix $P_{ij} = K_{ij}/(\sum_k K_{ik} q_k^{-1/2}) q_j^{1/2}, q_i = \sum_k K_{ik}$
7: Find $l$ largest eigenvalues $\gamma_k$ and eigenfunctions $\varphi_k$ of $P$
8: Form Galerkin matrices $A_{ij} = \langle \varphi_i, V\xi_j \rangle - \theta \langle \varphi_i, \Delta\xi_j \rangle, G_{ij} = \langle \varphi_i, \xi_j \rangle$
9: Solve $Ac = \lambda Gc$ for coefficients $c_k$ and eigenvalues $\lambda_k$
10: Compute eigenfunctions $\psi_i = \sum_{j=1}^l c_{ji} \varphi_j$
11: Calculate Dirichlet energies $E(\psi_i) = \langle \psi_i, \Delta\psi_i \rangle / \|\psi_i\|^2$
12: Order $(\lambda_k, \psi_k)$ by increasing $E(\psi_k)$
13: Compute frequencies $\omega_k = \text{Im}(\lambda_k)$

---

$$(P_{\tau/2} \mathcal{K} P_{\tau/2} \varphi_j)(x_n) \approx \int \tilde{k}_{\tau/2}(x_n, y) \varphi_j(y_{n+1}) d\mu(y), \tag{23}$$

where $\tilde{k}_{\tau/2}$ is the normalized kernel function associated with $P_{\tau/2}$. Thus, the entries of $\mathbf{A}_\tau$ become:

$$A_{\tau, ij} = \frac{1}{N} \sum_{n=1}^{N-1} \varphi_i(x_n) \int \tilde{k}_{\tau/2}(x_n, y) \varphi_j(y_{n+1}) d\mu(y). \tag{24}$$

Solving the generalized eigenvalue problem $\mathbf{A}_\tau c = \lambda \mathbf{G} c$, yields approximations of the eigenvalues $\lambda$ and eigenfunctions $\phi_\tau$ of the smoothed Koopman operator. This approach allows us to extract coherent dynamical patterns that are robust to perturbations and noise, while still capturing the essential features of the underlying dynamics.

**Computational Considerations** This approach enables efficient handling of large datasets through computational resource optimization. In Algorithm 1, the most computationally intensive operations comprise the kernel matrix $K_{ij}$ calculation (step 5) and the solution of two eigenvalue problems: one for $P$ (step 7) and another for the Galerkin solution (step 9). Through the implementation of sparse representations—specifically by moderating the nearest neighbors $k_{nn}$—and restricting the number of eigenfunctions $l$, we reduce the computational complexity of the eigenvalue problems while preserving the essential system dynamics. The effectiveness of working with a limited number of modes stems from the extracted eigenfunctions representing intrinsic dynamical patterns, allowing accurate system behavior capture with minimal modes. Furthermore, the kernel matrix $K_{ij}$ computation scales efficiently with data size through techniques such as random Fourier features (Giannakis et al., 2023). This computational framework achieves substantially faster training times compared to contemporary deep learning and neural network methods while maintaining robust performance in dynamic system characterization (Tavasoli et al., 2023).

## 3 RESULTS

In this section, we apply our spectral operator-based framework to live-cell imaging data to extract coherent temporal patterns in cellular dynamics. We begin by describing the dataset and performing a preliminary analysis to understand the divergence of cellular trajectories under different experimental conditions. We then demonstrate how our framework captures these dynamics and evaluate the representation performance in reconstructing and predicting ERK activity trajectories. The pseudo-code used to generate the Koopman results is reported in Algorithm 1.

**Datasets** The methodology was applied to a live-cell imaging dataset featuring retinal pigment epithelial (RPE) cells engineered with a doxycycline (DOX)-inducible $BRAF^{V600E}$ oncogene (Chen et al., 2023). The $BRAF^{V600E}$ mutation activates the mitogen-activated protein kinase (MAPK) signaling pathway, resulting in elevated extracellular signal-regulated kinase (ERK) activity, which regulates cell proliferation and differentiation.

The engineered cells expressed both the ERK activity reporter EKAREN5 and a cell cycle indicator (mCherry-dE2F PIP), enabling concurrent monitoring of ERK signaling dynamics and cell cycle progression. Live-cell imaging conducted at 10-minute intervals across four days captured the temporal evolution of ERK activity within individual cells.

To examine ERK inhibition effects on cellular dynamics, the experimental design incorporated varying concentrations of the ERK inhibitor SCH772984 (ERKi). The analysis focused on two experimental conditions:

1. DOX + Low ERKi: DOX-induced cells treated with low-concentration ERK inhibitor
2. DOX + High ERKi: DOX-induced cells treated with high-concentration ERK inhibitor

These experimental conditions facilitated investigation of ERK inhibition level effects on ERK activity dynamics and cellular responses.

**Kernel Operator and Block Structure** Using the delay-coordinate embedding with $Q = 5$ frames delays, we constructed the kernel matrix for the **DOX + Low ERKi** condition. The self-tuning kernel function captured the similarities between data points in the embedded space, and the resulting kernel matrix exhibited a distinct block-diagonal structure, as shown in Figure 1(above). The block-diagonal structure of the kernel matrix suggests the presence of distinct dynamical regimes or attractors in the cellular state space. This implies that cells transition between different states over time, and these transitions are captured by the coherent patterns in the data.

**Principal Koopman Modes** The extracted principal Koopman modes reveal dominant temporal patterns in ERK signaling dynamics at single-cell resolution. Figure 1(bottom) illustrates the first two Koopman modes obtained under the Low ERKi condition. The first Koopman mode characterizes a smooth temporal transition, indicating systematic state changes within individual cells. This transition pattern captures the progressive ERK activity suppression following ERK inhibitor introduction. The mode reveals the gradual shift from elevated to suppressed ERK activity states, consistent with established mechanisms of cellular signaling pathway adaptation to external perturbations (Eldar & Elowitz, 2010). This collective behavior demonstrates probabilistic state transitions in response to external signals—a characteristic feature of stochastic differentiation systems. The sec-

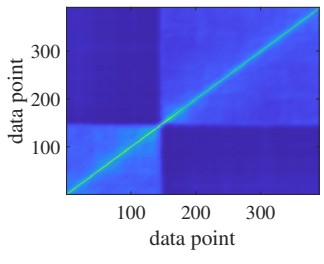

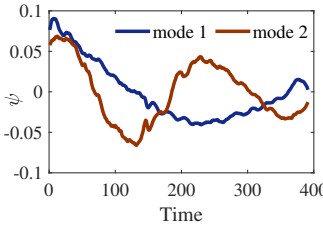

Figure 1: *(top) Kernel operator matrix showing block-diagonal structure, indicating distinct dynamical regimes. (bottom) The first two principal Koopman modes extracted from the data.*

ond Koopman mode exhibits periodic oscillations between positive and negative values. This pattern indicates intrinsic cyclical dynamics within the ERK signaling network, potentially arising from molecular fluctuations in low copy number species (Elowitz et al., 2002). These oscillations may correspond to cell cycle phases, regulatory feedback loops generating transient responses, and mRNA half-life effects that contribute to temporal variability in protein expression and signaling dynamics.

**Model ability to Reconstruct and Predict** Utilizing a sparse representation with only the 10 smoothest modes, we constructed a model to represent the individual cell ERK activity trajectories. This approach acknowledges the inherent stochasticity in cellular signaling by focusing on the most significant modes that capture essential dynamics while filtering out less predictable variations. We evaluated the model's performance on both the training set (Low ERKi condition) and unseen data—including data after frame 400 in the Low ERKi condition (with one frame every 10 minutes).

TRAINING SET PERFORMANCE    Figure 2 (the left plot) compares the model predictions with the observed ERK activity data for a **randomly chosen cell** under the Low ERKi condition. The model effectively captures the overall trends and key fluctuations in ERK activity, demonstrating a close match with the observed trajectory. This indicates that the dominant Koopman modes effectively encapsulate both the deterministic response to the inhibitor and the stochastic variations arising from intrinsic noise. By reconstructing the dynamics using a limited number of modes, the framework demonstrates its capacity to distill complex, noisy biological data into interpretable and predictive components.

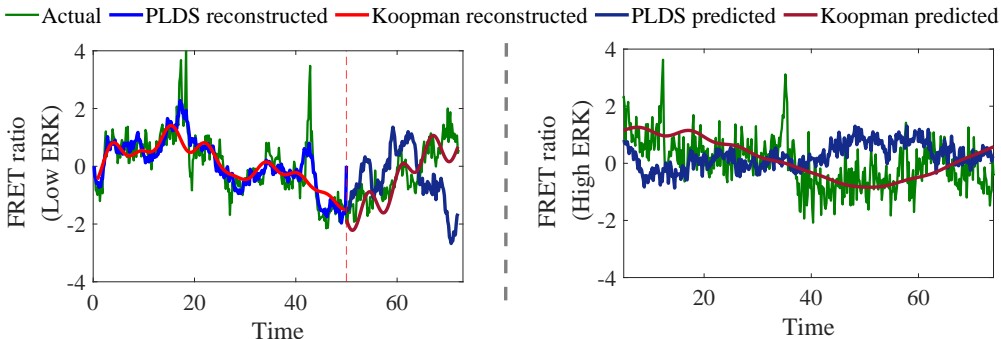

Figure 2: *Performance examples of model prediction for ERK activity trajectories in the Low ERKi condition (**left**, training set) and High ERKi condition (**right**, test set).*

GENERALIZATION TO UNSEEN DATA    We applied the model trained on the Low ERKi condition to the High ERKi condition without retraining, as shown in Figure 2 (right). The model predictions (red) align well with the observed data (green), capturing the general behavior of ERK activity under a higher level of inhibition. Despite the increased perturbation, the principal Koopman modes learned from the Low ERKi data remain relevant, suggesting that the fundamental dynamics of ERK signaling persist across different inhibition levels. This model transfer property is absent in modern approaches like PLDS, as illustrated in Figure 2.

This generalization implies that the Koopman eigenfunctions encapsulate conserved patterns in the cellular response, reflecting core mechanisms of how cells adapt to varying degrees of external stress. The persistence of these modes across conditions indicates that our framework effectively identifies the underlying structures governing the stochastic and nonlinear dynamics of ERK signaling. By capturing these essential features, the model enhances its applicability to various experimental conditions, offering a robust tool for understanding and predicting cellular behavior in response to different perturbations.

COMPARISON OF DIFFERENT METRICS    A comprehensive performance comparison of multiple metrics against contemporary approaches, specifically CODEX (Jacques et al., 2021) and PLDS (Chen et al., 2017), is presented in Table 1. The accurate prediction of cell dynamics requires capturing intricate temporal behaviors for understanding complex biological processes. Although CODEX demonstrates lower average error through effective population-level averaging, its limited representation of detailed dynamic variations reduces applicability in scenarios demanding high-fidelity transient behavior analysis. The inconsistency in CODEX performance becomes evident when examining predictions across LowERKi and HighERKi conditions in Table 1. While CODEX should theoretically achieve higher accuracy on the seen dataset (LowERKi) compared to the unseen dataset (HighERKi), the results contradict this expected pattern, raising concerns about methodological consistency.

The Koopman-based method demonstrates superior performance in capturing fine-grained dynamics, as illustrated through single-cell trajectories in Figure 2, enabling deeper insights into cellular behavior and enhanced predictive accuracy for precision-critical applications. The PLDS approach (Boots, 2009), implemented according to Chen et al. (2017), attempts reconstruction and forecasting

unlike CODEX, but exhibits limitations in transient capture due to inherent stability constraints, as evidenced in both numerical results and Figure 2. Additionally, while functional Principal Component Analysis (fPCA) represents a common analytical approach, its exclusion from Table 1 stems from inherent limitations in predictive capability beyond observed time periods.

Table 1: Performance Metrics comparison in heldout data for LowERKi and unseen data for High-ERKi Tests

| Metric | Koopman | | CODEX (Jacques et al., 2021) | | PLDS (Chen et al., 2017) | |
|---|---|---|---|---|---|---|
| | LowERKi | HighERKi | LowERKi | HighERKi | LowERKi | HighERKi |
| RMSE | 1.00(0.37) | 1.16(0.26) | 1.04(0.42) | 0.82(0.31) | 1.45(0.48) | 1.70(0.46) |
| MAE | 0.78(0.27) | 1.00(0.28) | 0.81(0.33) | 0.67(0.26) | 1.18(0.43) | 1.42(0.43) |
| MAPE (%) | 448(1742) | 382(465) | 672(2074) | 387(418) | 1233(4482) | 870(150) |
| R-squared | -1.08(3.20) | -4.31(6.06) | -1.16(2.68) | -1.70(4) | -7.50(24.34) | -11.44(15.39) |
| DTW Distance | 62(23) | 317(96) | 61(26) | 50(21) | 68(33) | 295(105) |

## 4 CONCLUSION

In this paper, we proposed a spectral operator-based framework that extracts coherent temporal patterns from live-cell trajectory data to characterize cellular responses to perturbations. The results demonstrate the existence of conserved temporal patterns within cellular dynamics, persisting through inherent biological stochasticity and variability. Our approach adopts a probabilistic representations and by approximating robust Koopman eigenfunctions, captures fundamental dynamical aspects that remain consistent across diverse external conditions, enabling deeper insights into complex biological processes. The framework presents notable advantages over functional data analysis and deep learning architectures. It generates interpretable representations through Koopman eigenfunctions that correspond to meaningful temporal patterns, contrasting with black-box model approaches. Furthermore, the framework demonstrates robust performance with limited variables measured in live-cell imaging, effectively addressing data constraints inherent to biological experiments. Through the integration of conserved dynamical pattern detection and stochasticity characterization, this approach advances the understanding of cellular decision-making and adaptation mechanisms.

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
