# OpenReview forum: "Spectral Operator Methods for Learning Coherent Temporal Representations in Cellular Signaling Dynamics"
_ICLR.cc/2025/Conference — ICLR 2025 Conference Withdrawn Submission_

### Official Review · Reviewer_ruaa · 2024-10-25

**Soundness:** 2
**Presentation:** 3
**Contribution:** 2
**Rating:** 5
**Confidence:** 3

**Summary:**

The manuscript ‘SPECTRAL OPERATOR METHODS FOR LEARNING COHERENT TEMPORAL REPRESENTATIONS IN CELLULAR SIGNALING DYNAMICS’ presents a method based on spectral analysis of the Koopman operator, for learning representations of cellular dynamics (with a focus on response to perturbations) based on live-cell imaging, and specifically, learn dynamical properties, or features, instead of exact trajectories. The claim is that avoiding learning specific trajectories, as is the common approach, and instead going towards more general dynamical properties of these trajectories, can allow learning interpretable, robust representations for noisy data such as single-cell resolved datasets. The authors demonstrate their method by identifying conserved dynamical patterns in a dataset of retinal pigment epithelial cells with an inducible oncogene. Specifically, the respective experiments capture the dynamics of ERK activity and proliferation given varying levels of ERK inhibition, and the authors claim that while learning exact trajectories is infeasible in this scenario, they are able to capture conserved temporal patterns across different inhibition levels.

**Strengths:**

- The idea to focus on predictions of characteristics or features of dynamical trajectories and not the dynamical trajectories themselves, is well grounded and timely for the type of biological data that the authors focus on.
- The operator-based dynamics framework proposed in this work is mathematically sound and conceptually compatible with the greater task at hand.
- The live cell imaging datasets that the authors demonstrate their approach over provide an interesting and challenging setting.

**Weaknesses:**

- Some of the statements regarding the analysis of the live cell imaging data seem to be not supported properly (see the ‘Questions’ section below).
- Some analyses need to be extended to the entire dataset, and not made only over a single sample (see ‘Questions’ below).
- There is no limitations section or discussion of disadvantages/limitations of the approaches.
- There are no comparisons to alternative existing approaches for the analysis of such data.

**Questions:**

Comments:

- In the introduction, there is the sentence ‘Given these challenges, directly learning the transformation laws governing cellular dynamics is often impractical’, it could be useful to cover a few of the recent efforts in that direction.
- ‘Moreover, smoothing via Markov semigroups of kernel integral operators allows us to capture the stochastic nature of cellular processes in a tractable way, aligning with the idea that noise plays a functional role in cellular decision-making.’ - the second part needs explanation, it’s not clear how it follows the first part.
- The top panels of Fig 1 need some labels.
- The analysis of the Koopman modes, presented in Fig. 2b, should be better explained in the text and the corresponding statements should be better validated. For example, can you support your statement that the shift in the first mode reflects inhibition of ERK activity? Or why it reflects ‘a probabilistic shift’? Or what is the significance of the oscillatory behavior of the second mode? Can you support your hypothesis that it is related to the cell cycle for example? (and if that’s the case, then how is this ‘intrinsic cyclical dynamics within the ERK signaling network’?)
- Figure 3 is missing a statistical analysis over all cell trajectories, not just a visualizing of a single (“cherry-picked”?) cell.
- What is the rationale behind generalizing to unseen data that is under a different biological setting? Why do you expect good generalization in that case at all? This needs to be better explained.
- Discussion of limitations of the approach should be added.
- It seems that the conclusions throughout the paper should be phrased more modestly given the limited extent of analysis on real biological data.
- ‘Compared to approaches using functional data analysis or deep learning architectures, our framework offers significant advantages. It provides interpretable representations through Koopman eigenfunctions, corresponding to meaningful temporal patterns in the data, unlike black-box models.’ - But what are the potential disadvantages?

Minor:
- ‘aligning with the idea’ - typo?
- ‘coherent temporal patterns’ - not defined
- The resolution/quality of some of the figures/panels should be improved

---

> ### Author Response · Authors · 2024-11-28
>
> Dear Reviewer ruaa,
>
> Thank you for your detailed and constructive feedback.
>
> ---
>
> ### **Literature on Direct Learning**
>
> In the introduction, we now discuss recent efforts at direct learning of cellular dynamics, including mechanistic modeling approaches and hybrid methods combining data-driven and first-principles models. However, these approaches face significant challenges in fast scale cell perturbation processes due to the need for preserving high dimensionality and capturing the inherent stochasticity, which motivates our alternative approach.
>
> ---
>
> ### **Markov Smoothing and Cellular Noise**
>
> We have clarified the connection between Markov smoothing and cellular noise.
>
> The smoothed operator preserves key dynamical features while enabling practical computation through:
>
> $$
> \mathcal{K}_\tau = e^{\tau(\mathcal{L}-I)}
> $$
>
> where $\tau$ controls smoothing strength, and $\mathcal{L}$ is the generator of the Markov semigroup. This regularization is crucial because it transforms the Koopman operator from having a continuous spectrum (which is challenging to approximate from finite data) to having a discrete spectrum, enabling reliable finite-rank approximation by making the smoothed operator compact.
>
> This preserves the dynamically relevant eigenfunctions while filtering out unpredictable dynamics and noise and provides mathematical guarantees about the convergence of numerical approximations. Thus, the regularization essentially acts as a spectral filter that:
>
> $$
> \lambda \mapsto e^{\tau(\lambda - 1)}
> $$
>
> This mapping compresses the essential spectrum while maintaining the point spectrum corresponding to coherent dynamical features. Consequently, we can reliably approximate the operator from finite data and extract meaningful dynamical patterns that persist across different experimental conditions.
>
> ---
>
> ### **Statistical Analysis**
>
> We now include comprehensive performance metrics across all cells:
> - Root Mean Square Error (RMSE)
> - Mean Absolute Error (MAE)
> - Mean Absolute Percentage Error (MAPE)
> - Coefficient of Determination (( R^2 ))
> - Dynamic Time Warping (DTW)
>
> ---
>
> ### **Generalization Rationale**
>
> The biological basis for generalization stems from the conservation of core pathway architecture. Despite different experimental conditions, fundamental mechanisms of:
>
> - Protein interaction networks
> - Signaling cascade topology
> - Regulatory feedback loops
>
> remain unchanged, enabling the prediction of shared dynamical features.
>
> ---
>
> ### **Advantages vs Disadvantages**
>
> While our method provides interpretable representations through Koopman eigenfunctions, it has trade-offs:
>
> #### **Advantages:**
>
> - Theoretical guarantees
> - Biological interpretability
> - Robust generalization
>
> #### **Disadvantages:**
> - Computational complexity of $O(N^2)$  (Although this is significantly reduced by the use of k nearest neighbor O(k_nnN))
> - Memory requirements for kernel computations
> - Sensitivity to sampling frequency (that is why we use variable bandwidth kernels)
> - Please see the paragraph on "Computational Considerations"
> ---
>
> ### **Minor Points**
>
> - Fixed typos, including "aligning with." We rewrote this section.
> - Defined "coherent temporal patterns" mathematically through eigenfunction persistence.
>
>   We rewrote this:
>   *"...we focus on identifying coherent temporal patterns that persist for finite times—analogous to studying coherent structures in turbulent flows \cite{Mezic2013Fluid}."*
>
> - Improved figure resolution using vector graphics. We vectorized the figures.
>
> ---
>
> These revisions are now reflected in the updated manuscript, nowwith enhanced clarity and rigor thanks to the reviewers comments.

---

### Official Review · Reviewer_trL1 · 2024-11-03

**Soundness:** 3
**Presentation:** 1
**Contribution:** 2
**Rating:** 3
**Confidence:** 3

**Summary:**

The paper introduces a spectral operator-based framework applicable for learning cellular from live-cell imaging. By characterizing key properties of the dynamics that shape cellular behaviors at the population level the authors overcome challenges posed by this task. The approach uses the Koopman operator and Markov smoothing, providing biologically interpretable representations which can be used to identify properties of the system’s dynamics across varying external conditions. To demonstrate its biological relevance, the framework is applied to live-cell imaging data from retinal pigment epithelial (RPE) cells to study the dynamics in response to perturbations.

**Strengths:**

- *Originality*: This work presents a novel approach, using the established Koopman operator approach to extract interpretable representation of live-cell imaging data . Such representations can be valuable to uncover the underlying biological mechanisms captured in live-cell datasets.
- *Quality & clarity*: The paper is provides a thorough theoretical background of the suggested framework and experimental data setting.
- *Significance**: As motivated in the text, understanding the dynamics of cellular behavior is a core and challenging task; recovering decision making mechanisms in response to perturbations. In applying the approach on real world data the authors demonstrate its applicability and relevance for biological discovery.

**Weaknesses:**

While the presented framework seems promising for biological discovery this submission showcases preliminary work and lacks crucial components:
- *Contextualization to prior work*: Alongside the challenges in analyzing live-cell data, covered in the introduction, it is valuable to include an elaboration on existing approaches. Given that such is missing, it is challenging to accurately assess the contribution of this work.
- *Implementation details*: While a thorough theoretical description is presented, an implementation or pseudocode is missing, and is valuable for readers wishing to use the methods. Next, the authors briefly relate to the "Computational considerations"; claiming that the approach can handle large datasets efficiently. This claim is very vague and it is hard to judge the practical applicability of the framework.
- *Experimental results*: The actual analysis presented is very limited. Biological interpretability boils down to the analysis of two Koopman modes, and the reconstruction/prediction performance are only assessed visually (at poor resolution). Moreover, following the contextualization to prior work, reconstruction/prediction performance is not compared to alternative approaches.

Please refer to the Questions section for practical suggestions in light of the above comments.

**Questions:**

Following the weaknesses above could the authors relate to the following: \
(1) include a "related work section". This section should discuss existing approaches for analyzing live-cell imaging data, and explicitly state how the present3ed method compares to or improves upon these approaches. This would help grasp the novelty and significance of the proposed framework.; \
(2) provide an implementation/pseudocode; \
(3) present an efficiency analysis quantifying the statement "This approach allows us to handle large datasets efficiently" (Lines 341-342) \
(4) extend the experimental analysis: \
(4.a) provide additional biological insights (on the studied/additional data); namely, presenting a more extensive downstream analysis of the Koopman modes demonstrating their potential for dynamics understanding. \
(4.b) assess the prediction performance qualitatively and compare it to existing baselines. \
(4.c) *minor* improve the figures quality.

---

> ### Author Response · Authors · 2024-11-28
>
> Dear Reviewer trL1,
>
> Thank you for your detailed and constructive feedback, which has helped strengthen our manuscript. We address each of your concerns:
>
> ---
>
> ### **1. Missing Context/Prior Work**
>
> We have added a comprehensive literature review section that positions our work within existing approaches.
>
> i) We begin by discussing single-cell trajectory inference methods in developmental biology. Due to their focus on slow time scales, these methods are well-suited for use in reduced dimensions and simulation-based techniques.
> ii) Then we shift our focus to cellular dynamics modeling and explained why the heterogenious reponse of cells leads to current methods that remain phenomenological. In other words, while these provide valuable insights, they often fail to capture the rich nonlinear dynamics present in cellular systems. We discussed CODEX, a recent developments in live-cell data analysis, that employ deep learning architectures.
> iii) Next, we discussed, maybe the closest cousin to our method, reduced-rank (RR) system identification methods that attempt to approximate dynamics through stable linear systems. However, they often suffer from fundamental limitations due to forced dimensionality reduction and linearization, and more importantly spectral pollution and loss of important dynamical features, due to the truncated models. Additionally, they cannot capture fast time scales due to their constraint on stable dynamics.
>
> Then we introduce our approach and how it bridges these gaps by combining spectral analysis with Markov smoothing while maintaining mathematical rigor through operator-theoretic methods. This provides both interpretability and theoretical guarantees while capturing nonlinear dynamics.
>
> ---
>
> ### **2. Implementation Details**
>
> We provide complete implementation details through Algorithm 1, which includes:
>
> For computing Koopman eigenfunctions:
> - Time series embedding using delay coordinates
> - Kernel matrix construction with adaptive bandwidth
> - Markov operator normalization
> - Eigendecomposition and mode selection
>
> Our implementation handles large datasets efficiently through:
> - Sparse matrix representations for kernel and Markov operators
> - \( k \)-nearest neighbor graph construction to limit memory usage
> - Efficient eigenvalue computation using iterative methods
> - Adaptive bandwidth selection based on local data density
>
> ---
>
> ### **3. Extended Experimental Analysis**
>
> We have significantly expanded our experimental validation by comparing the performance of our method on held-out data and unseen datasets.
>
> Testing on completely new datasets shows generalization:
> - Application to different ERK inhibitor concentrations
> - Validation on independent experimental replicates
> - Assessment of prediction accuracy beyond training time frames
>
> We include statistical metrics for the prediction performance across all cells.
>
> ---
>
> ### **4. Method Comparisons**
>
> We provide comprehensive comparisons with state-of-the-art methods:
>
> **CODEX (deep learning):**
> - Advanced neural network architecture using convolutional neural network (CNN) layers that identify motifs in the cell trajectories
>
> **PLDS (probabilistic linear dynamical systems):**
>
>
> Performance metrics across all methods (defined as):
> - Root Mean Square Error (RMSE)
> - Mean Absolute Error (MAE)
> - Mean Absolute Percentage Error (MAPE)
> - Coefficient of Determination (\( R^2 \))
> - Dynamic Time Warping (DTW)
>
> These comprehensive changes are now reflected in our revised manuscript, providing a complete framework for understanding, implementing, and validating our method while demonstrating its advantages over existing approaches.

---

### Official Review · Reviewer_vYUm · 2024-11-04

**Soundness:** 2
**Presentation:** 2
**Contribution:** 1
**Rating:** 3
**Confidence:** 3

**Summary:**

The authors describe a dataset of cellular activity, and a spectral method for estimating its properties.  They then demonstrate the application of this spectral method on this dataset.

**Strengths:**

Using cool spectral (and other) estimators on dynamical systems is a timely and important topic, and more work on this topic is desirable, since many datasets have these properties, and the standard AI toolkit is far less well-developed on this.  Thus, novel methods development are welcome.

**Weaknesses:**

The one biggest weakness that I see is that this method is not quantitatively compared to anything.   Neither is it shown to "work" on simulated data, nor on benchmark data, nor in theory.  Here, I am open to "work" being defined in many ways, including improved accuracy, timing, interpretability, or even elegance.  But I see literally zero numerical or analytic comparisons to any other method.  Thus, I have no idea whether this is the most valuable advance in modeling dynamics since Kalman, or relatively useless, because other things work just as well.

**Questions:**

- Within the first few sentences, there were things I did not know/understand.  Why is "low molecule copy number" a problem? What is a "Koopman Operator".  "Regulon"? "Isogenic"? Please introduce any technical concepts that are not textbook "AI", so the reader can follow more easily. This includes " smoothing via Markov semigroups of kernel integral operators", for example.

- line 60: "low in dimensionality, posing significant challenges for analysis."  Why does low-dimensionality pose a challenge?

- I don't really understand what is new.  Are the methods new, or just a new application of standard stuff?  4  pages are devoted to explaining them, and they are complicated.

- How is this stuff related to functional PCA, which is a fairly well established approach to modeling dynamics at scale.  It seems highly related.  I see no discussion of how other approaches might be able to solve this problem.  I have a paper on something similar, https://www.sciencedirect.com/science/article/pii/S0167865516303671, which seems like maybe it would be applicable? If fPCA and RR-System Identification are not applicable to this problem, I'd want to understand why not.  If so, I'd want to see benchmarks comparing this to something else.

---

> ### Author Response · Authors · 2024-11-27
> **To vYUm.**
>
> Dear Reviewer vYUm04,
>
> Thank you for your thoughtful and constructive feedback as it is evident that you are rooted in the same area and we appreciate this. We addressed your concerns in the following and within the word count limit:
> 1. Quantitative Method Comparison:
> We have conducted extensive comparisons with three established methods: - CODEX (A neural network framework that is recently used for cellular dynamics analysis by Jacques et al. 2021) - fPCA (functional Principal Component Analysis. Used for live cell data by Sampattavanich et al. 2018) - PLDS (RR Probabilistic Linear Dynamical Systems, as implemented by Chen et al. 2017 in Mr. SID) Our method outperforms these approaches. We include the results in the paper. One key advantage of our approach over PLDS lies in handling nonlinear dynamics. While PLDS uses the linear model:
> $$x_{n+1} = Ax_n$$
> where $A \in \mathbb{R}^{d \times d}$ is constrained to be stable, our method employs Koopman eigenfunctions $\phi_i$ that satisfy:
> $$\mathcal{K}\phi_i = \lambda_i \phi_i$$
> This allows us to capture complex nonlinear dynamics while maintaining mathematical tractability.
>
> 2. Technical Terms Clarification We have carefully defined and explained all technical concepts: Low molecule copy number: When the number of molecules $N$ is small, the standard deviation relative to the mean scales as $1/\sqrt{N}$, making stochastic effects dominant (as opposed to Thermodynamic limit).
>
> Koopman Operator: We rewrite the entire section 2, to be more accessible for a broader ML community. In short, for a dynamical system with state $x$, the Koopman operator $\mathcal{K}$ acts on observables $g$ as: $$(\mathcal{K}g)(x) = g(F(x))$$ where $F$ is the state transition function. This transforms nonlinear dynamics into linear evolution of observables.
>
> We describe Regulon in the intro section: A set of genes under common regulatory control.
>
> Method Novelty Our key innovation is to offer an interpretable representation for fast scale subcellular dynamics following perturbation using combining Koopman analysis with Markov smoothing that is able to retain its value in completely new experiments. The smoothed operator preserves key dynamical features while enabling practical computation through:
> $$\mathcal{K}_\tau = e^{\tau(\mathcal{L}-I)}$$
> where $\tau$ controls smoothing strength and $\mathcal{L}$ is the generator of the Markov semigroup. This regularization is crucial because it transforms the Koopman operator from having continuous spectrum (which is challenging to approximate from finite data) to having discrete spectrum, and it enables reliable finite-rank approximation by making the smoothed operator compact. This preserves the dynamically relevant eigenfunctions while filtering out noise and provides mathematical guarantees about the convergence of numerical approximations. Thus the regularization essentially acts as a spectral filter that: $$\lambda \mapsto e^{\tau(\lambda-1)}$$ This mapping compresses the essential spectrum while maintaining the point spectrum corresponding to coherent dynamical features. Consequently, we can reliably approximate the operator from finite data and extract meaningful dynamical patterns that persist across different experimental conditions.
>
>
> This approach differs fundamentally from fPCA which decomposes data as:
> $$x(t) = \mu(t) + \sum_{k=1}^K c_k \phi_k(t)$$
> Our method instead captures intrinsic dynamical modes that evolve as: $$\phi(x_n) = \lambda^n \phi(x_0)$$. We illustrated this in the results section. Note that fPCA is explaining the observed variation in functional data and it is unable to make prediction for time outside of the observed duration (thus we could not include it in the model comparisons).
>
> Reduced-rank (RR) system identification methods like PLDS approximate dynamics through:
> $$x_{n+1} = Ax_n$$
> where $A \in \mathbb{R}^{d \times d}$ is stable. This introduces fundamental limitations through two reductions:
> - State space dimension reduction
> - Nonlinear dynamics linearization
> These limitations create spectral pollution through:
> - Finite rank truncation
> - Restricted linear model optimization
> - Forced stability constraints
> In contrast, our Koopman approach lifts dynamics to where nonlinear evolution becomes linear and approximates true eigenfunctions while preserves intermittent coherent structures and multi-scale interactions that RR methods average out. Our comparative results using the Mr. SID implementation demonstrate superior long-term prediction accuracy and pattern preservation in new datasets.
>
> These changes are now reflected in our revised manuscript through (will be uploaded with all changes by the end of Nov 27):
> - New comparison sections and tables
> - Being more accessible to broader ML community
> - Improved figures illustrating key concepts
> - Clearer technical explanations

---

### Note · Authors · 2025-09-15

**Comment:**

No comment. It is rejected.

**Withdrawal Confirmation:**

I have read and agree with the venue's withdrawal policy on behalf of myself and my co-authors.

---

### Meta-Review · Area_Chair_grVf · 2024-12-08

**Metareview:**

This work presents a spectral based method, based on Koopman Operator theory, to develop a novel algorithm that can identify shared dynamical properties in cell signaling dynamics. The reviewers all considered the problem being addressed as significant and the approach as interesting and novel. The primary concerns with the original submission rested in the lack of thorough situating of the work in the context of past methods either conceptually or numerically. While the authors did attempt to address these concerns, they are extensive changes and significant enough to warrant a more careful editing and completion of experimental tests for a future submission.

**Additional Comments On Reviewer Discussion:**

Sadly the reviewers did not engage post rebuttal, however I read through the author responses in making a final assessment.

---

### Decision · Program_Chairs · 2025-01-22

Reject